# Bearing Thickness Is Not a Predictive Factor for Damage and Penetration in Oxford Unicompartmental Knee Arthroplasty—A Retrieval Analysis

**DOI:** 10.3390/ma13204589

**Published:** 2020-10-15

**Authors:** Johannes Adrian Eckert, Ulrike Mueller, Tilman Walker, Martin Schwarze, Sebastian Jaeger, Jan Philippe Kretzer

**Affiliations:** Laboratory of Biomechanics and Implant Research, Clinic for Orthopedics and Trauma Surgery, Heidelberg University Hospital, Schlierbacher Landstraße 200a, 69118 Heidelberg, Germany; Johannes.Eckert@med.uni-heidelberg.de (J.A.E.); ulmuel@hotmail.de (U.M.); Tilman.Walker@med.uni-heidelberg.de (T.W.); Martin.Schwarze@med.uni-heidelberg.de (M.S.); Sebastian.Jaeger@med.uni-heidelberg.de (S.J.)

**Keywords:** oxford unicompartmental knee arthroplasty, bearing thickness, retrieval analysis

## Abstract

The medial Oxford unicompartmental knee arthroplasty (OUKA) shows good survivorship, as well as clinical results. Aseptic loosening, however, remains one of the main reasons for revision and polyethylene debris is known to cause aseptic loosening. The role of bearing thickness in total as well as unicondylar knee arthroplasty has been the subject of controversial discussions, especially the longevity of lower thickness bearings in total knee arthroplasty was questioned. The purpose of this study was to assess the influence of bearing thickness on time to revision, damage pattern, penetration, and volumetric material loss. A cohort of 47 consecutively retrieved medial OUKA bearings was analyzed with conventional direct light microscopy applying the Hood damage analysis, as well as measuring the penetration depth. In this retrieval cohort, a difference on survival time, damage, penetration, as well as volumetric material loss could not be seen. We conclude that low as well as high thickness bearings can safely be used in OUKA without any relevant differences in terms of wear and damage.

## 1. Introduction

The Oxford unicompartmental knee arthroplasty (OUKA) has been shown to generate good clinical results. Due to the increased use of highly crosslinked polyethylene (HXLPE), the number of revisions due to wear has decreased [1,2,3] (the Annual Report of the Australian National Joint Registry notes only 1.4% of direct tibial insert wear [4]). However, aseptic loosening—which can be caused by wear—is still the major cause for revisions of UKA [4,5], the 15-year revision rate is around 11% [6]. The use of a spherical femoral component and a fully congruous meniscal bearing to increase the contact area theoretically reduces the potential for polyethylene wear [7]. Mobile bearing unicondylar designs, however, can exhibit wear at the femoral, as well as the tibial aspect of the inlay [8,9]. Especially in cases of suboptimal bearing positioning, the wear can increase drastically [10]. To date, there is no scientific consent as to which inlay thicknesses generate better outcomes. Longer times to revision in Oxford UKA have been described for 3 and 4 mm bearings, respectively, compared to thicker bearings [10]. In addition, the use of thinner inlays allows for a reduced loss of bonestock and usage of thicker bearings has a higher risk for overcorrection of the mechanical axis, resulting in a higher post-operative valgus [11]. In incongruent articulations, however, the wear rate of polyethylene is increased when it is used in a thin layer [12,13], and bearing failures have been described for inlays <6 mm, caused by wear [14,15]. Accordingly, the FDA has issued a Class II Special Controls Guidance Document in 2003, recommending a minimum thickness of 6 mm at the lowest point below the condyle (or alternatively the demonstration of the ability of the said bearing to survive 10 million cycles in a physiological knee simulator). Unfortunately, the information provided by manufacturers with regards to the actual inlay thickness often appears to be inaccurate or misleading [16].

In this retrieval analysis of a series of 47 consecutively explanted medial Oxford unicondylar knee replacements, we assessed the types of damage using the modified Hood score and analyzed penetration depth and volumetric changes during the lifetime of the retrieved bearings [17]. We wanted to assess whether the use of higher-thickness inlays (5–6 mm in this cohort) leads to higher wear than the use of low-thickness inlays (3–4 mm) and if different types of damage are seen in the two groups.

## 2. Materials and Methods

From October 2013 to August 2018, 47 ultra-high molecular weight polyethylene (UHMWPE) meniscal bearings of the medial Oxford unicompartmental knee arthroplasty (OUKA) from 46 patients were consecutively explanted and included in this retrieval analysis. All retrievals were conducted at our hospital. Of the 47 inlays, the majority (*n* = 41) was of the newer ’anatomic’ generation, only six were old type inlays. Eight of the inlays were small (all anatomic), 19 were medium-sized (16 anatomic, three non-anatomic), and 20 were large (17 anatomic, three non-anatomic). No XS inlays were explanted in the period examined. Of the revised knees, 21 were left knees, 26 were right knees. In 38 cases, the primary procedure was performed in our hospital, the remaining eight procedures were performed elsewhere. Primary implantations were conducted from October 2002 to September 2017. The manufacturer changed the type of UHMWPE in 2005. In this cohort, five of the retrieved inlays were implanted before, the rest after the change in material. The UHMWPE used before 2005 was Hi-Fax 1900, the PE currently in use is GUR 1050. A publication by Mohammad et al. showed no significant differences in wear between the two types of UHMWPE using a RSA analysis [18]. Microscopical images of the types of damage were made using a Keyence VHX-5000 reflected-light microscope.

The reasons for revision in descending order were progression of disease in other compartments (*n* = 19; 30%), followed by aseptic loosening (*n* = 9; 14%) and bearing dislocation (*n* = 5; 8%). Other reasons for revision were periprosthetic fractures, persistent postoperative pain, instability, infection, etc.

Following an institutional review board (IRB)-approved protocol, all orthopedic devices were collected, cleaned, and cataloged in an electronic database for long-term storage in an inert environment.

Surface damage of the retrieved meniscal bearings was assessed using light stereomicroscope analysis at magnifications from ×10 to ×20. The tibial bearings were divided into three regions (front-anterior, front-posterior, and back; see Figure 1). For these regions, the wear was quantified using the modified Hood-method [17], a method which has since been used for most macroscopic retrieval analyses [19,20]. Eight types of damage were assessed: Burnishing, scratching, pitting, delamination, surface deformation, abrasion, third-body embedded debris, and edge loading (Figure 2). Grading was performed by two graders (JAE, JPK) blinded to the clinical and radiographic data. Scores of 1, 2, and 3 indicated damage areas of <10%, 10% to 50%, and >50% of the surface area, respectively. The total score for each bearing was the sum of the eight individual damage scores over the three zones. The maximum score for each surface was 72.

The thicknesses of the tibial bearings as specified by the manufacturer for the current implant design of the OUKA range from 3 to 9 mm. The thicknesses of the retrieved bearings in this study ranged from 3 to 6 mm, 3 and 4 mm inserts were classified as low thickness inserts, 5 and 6 mm were classified as high thickness inserts. The manufacturer does not further specify the thicknesses, however, the nominal thickness of each bearing is always bearing thickness +0.5 mm. The mean variance of +0.130 mm was added, as the nominal thickness does not resemble the true thickness of a new inlay. We have found a mean variance of +0.130 mm in 10 brand new inlays, measured for this purpose.

To establish the linear penetration, the minimum thickness of each retrieved bearing was subtracted from the initial bearing thickness. The thickness was measured using a digital dial gauge (MarCator 1075 R, Mahr, Göttingen, Germany). A similar dial-gauge technique was reported by Psychoyios et al. to determine the minimum thickness of similar bearings [21]. Each bearing was mounted on a precision comparator stand with a universal measuring arm and a probe with a spherical tip of 4 mm in diameter was used. The dial gauge was determined to have a mean error of 0.0005 mm. Each bearing was measured once by two independent observers (JAE and JPK) and the mean of these values was taken as the minimum thickness of the bearing surface. Linear penetration was then calculated using the following formula:

Linear penetration (LP) (mm) = TN + TC − TR

TN = Nominal thickness = Bearing size + 0.5 mm

TC = Tolerance correction thickness = 0.13 mm

TR = Thickness at retrieval = Measured thickness (mm)

As the inlay is a fully congruent bearing, covered in the backside base area by the tibial plate and in the top area by the femoral component, the approximated volumetric wear rate was additionally assessed as follows:

Approximated volumetric wear rate (AWR) = ((base area + top area)/2) × LP.

The base areas were established dependent on the size and the design (anatomic/non-anatomic) of the bearing. Due to the necessity of a correction factor, especially in short time periods to revision, a negative approximated volumetric change—meaning an increase in thickness of the retrieved bearing—can occur. This was the case for *n* = 4 bearings (IDs 126, 199, 268, 1168). Four other inlays (IDs 92, 100, 675, 1092) had a time to revision <1 month. These eight bearings were excluded from the analysis of the penetration rate, as well as the approximated volumetric change.

## 3. Results

The data collected showed a comparable result between the two independent examiners (R = 0.82). The predominant type of damage was burnishing, followed by scratching and pitting. Edge loading, abrasion, and surface deformation were less common, while delamination and embedded particles were seen very rarely (Figure 3). Burnishing showed the highest number for all portions of the bearing, scratching—while seen in every specimen—was equally distributed over the Hood score. Pitting was also evenly distributed, yet some inlays did not exhibit any signs of pitting.

An ANOVA was conducted to compare damages for the four different bearing sizes. The mean total damage scores did not differ significantly in the four groups in this cohort (p = 0.86; Figure 4).

The lowest damage score could be seen at the backside of the implant, while front-anterior and front-posterior showed comparable results. As expected, the total damage score increased with the ongoing time to revision (Figure 5), although there was only a weak correlation (R = 0.3; R² = 0.09, *p* < 0.05).

The linear wear of the inlay increased with time (Figure 6).

A Shapiro-Wilk test was performed to check for normal distribution with a p > 0.05. The Shapiro-Wilk test revealed non-normal distribution and a Wilcoxon signed-rank test was performed for the subsequent analyses. The low-thickness inlays showed a mean penetration of 37.5 µm per year, whereas the high-thickness inlays exhibited lower penetration of 27.6 µm per year, the difference was not significant (Figure 7, p = 0.173).

The total damage score and the linear penetration did not show any correlation (R = 0.137; p = 0.383). The approximated wear volume for all inlays increased over time (R = 0.53, p = 0.001). The mean approximated wear volume for low-thickness inlays was 26.3 mm³ per year (R = 0.59, p = 0.001) and for high-thickness inlays 18.0 mm³ per year (0.53, p = 0.001) without significant differences (Figure 8, p = 0.196).

Finally, there was no significant difference in survival time for the two different groups (Figure 9, p = 0.264).

## 4. Discussion

In this retrieval analysis, the most common damage forms in descending order were burnishing, scratching, and pitting. This is consistent with earlier studies showing the same pattern of damage in total knee replacement [22,23]. However, also different damage patterns have been described and the pattern seems to be strongly influenced by the type of bearing [24,25]. In fully congruent meniscal bearing, extreme forms of damage such as edge loading are only expected to occur in small subsets of specimen, as they are rarely seen in cases of dislocation of subluxation of the inlays [26], while others such as embedded particles are also associated with the implantation technique. In this cohort, most types of damage distributed equally between the different surfaces (anterior, posterior, and back—i.e., tibial articulating surface). Abrasion was rarely seen at the backside which is to be expected, given the fully congruent nature of the tibial articulating portion of the bearing. In UKA as well as TKA, it has been advised against the use of thin inlays, due to the higher risk of thinning, increased wear, and risk for revision favoring the use of thicker inlays [27,28]. However, in our study we could find no such connections. While the linear penetration and the approximated volumetric wear rates were higher for the low thickness bearings, these findings were insignificant. Likewise, the survival rates for both groups showed comparable results. The total damage assessed by the Hood score and the penetration per month did not show any correlation. While counter-intuitive, this has been described by other groups before [29,30].

The penetration of 37.5 µm per year for low thickness and 27.6 µm per year for high thickness bearings is in range with studies published previously. Engh et al. described rates of approximately 50–60 µm per year for the medial and 30 µm for the lateral compartment in a type of total knee replacement [22], which is in line with results reported by Kop and Swarts who reported 54 µm for the superior surface of LCS knees [31]. For the Oxford unicompartmental knee arthroplasty, Horsager et al. described wear rates of 40 and 50 µm per year for the cemented and cementless versions, respectively, which is slightly lower than what we have seen in our cohort [32]. The aforementioned study was conducted using the radiostereometic analysis (RSA), which has been shown to produce accurate results in total knee replacement wear analysis [33,34]. However, no study comparing wear assessed by RSA and subsequently by the retrieval analysis has been conducted so far, to the best of our knowledge, and a certain error in measurement has to be accepted. In the present study, a direct measurement of the retrieved inlays was possible, which can be assumed as the most valid form to measure inlay penetration.

There are several limitations to our study: Since not all of the prostheses were implanted in our hospital, we could not assess clinical and radiological data. We analyzed both anatomical and non-anatomical bearings and the polyethylene used in the different generations of bearings does differ. Furthermore, as is the case for most retrieval studies, this was conducted as a retrospective analysis. Accordingly, the radiological examinations were not standardized to the need of the analyzer but to the need of the clinician. Hence, we could not properly assess pre- and post-operative valgus/varus, which might have an influence on the types and extent of damage. Furthermore, a correlation between patient satisfaction and experimental findings could not be investigated. Finally, as knee revision surgery is a complex and rarely standardized procedure, damages to the bearing and the rest of the implant can occur during the aforementioned surgery, although the differences between surgically induced damage and damage through wear and tear can be distinguished quite easily as done in this study.

To the best of our knowledge, this is the largest independent series of consecutively retrieved bearings of the medial Oxford unicompartmental knee arthroplasty to date. We could not find any significant differences in the two analyzed groups (high thickness—5 and 6 mm—and low thickness—3 and 4 mm—inlays) with regards to damage score, types of damage, penetration, approximated wear rate, and survival time.

## 5. Conclusions

In this retrospective retrieval analysis, we wanted to assess whether bearing thickness influences the clinical outcome and biomechanical aspects, with a special focus on time to revision and wear. We could not find significant differences for low- and high-thickness inlays. This study suggests that inlays with the analyzed bearing thicknesses can safely be used in OUKA without an increased risk for a decreased time to revision due to higher wear.

## Figures and Tables

**Figure 1 materials-13-04589-f001:**
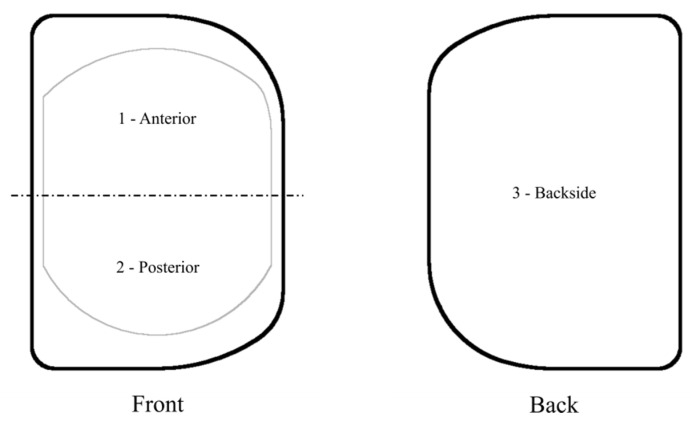
The front and back of the meniscal bearings were divided into three sections. Surface damage grading scores were assigned for each section using light stereoscopic examination and a subjective grading score.

**Figure 2 materials-13-04589-f002:**
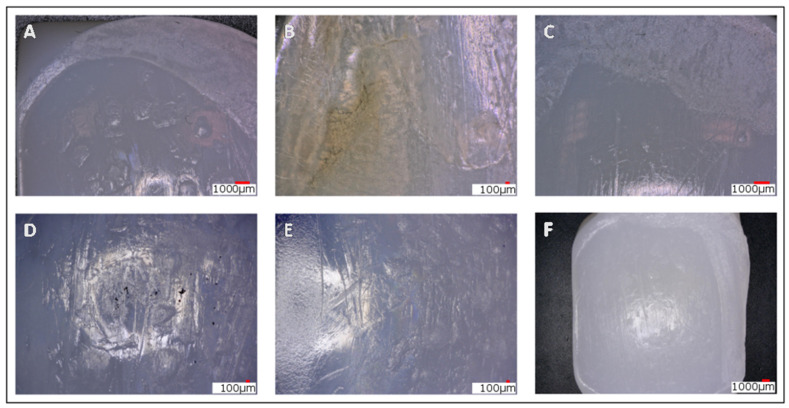
(**A**) Edge loading and abrasion in the anterior medial border of the inlay, pitting throughout the inlay, small scratches; (**B**) delamination at the anterior medial border of the inlay (noticeable at the brown change in color little pitting and scratching; (**C**) abrasion, scratching, and little pitting; (**D**) embedded particles (small black dots, possibly metal debris), a lot of scratching and pitting; (**E**) burnishing (see changes of regular manufacturing marks on the left side to a polished surface); (**F**) surface deformation, edge loading, abrasion, and a little pitting and scratching.

**Figure 3 materials-13-04589-f003:**
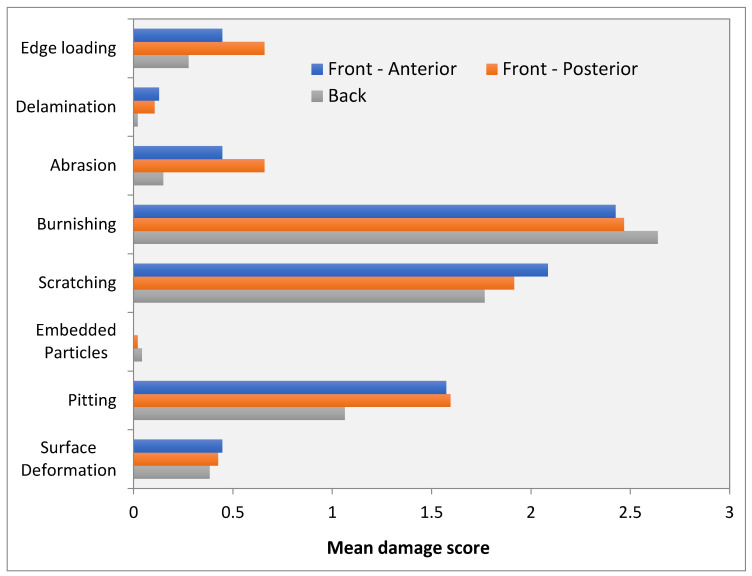
Hood damage score for all parts of the inlay (front-anterior and -posterior; back).

**Figure 4 materials-13-04589-f004:**
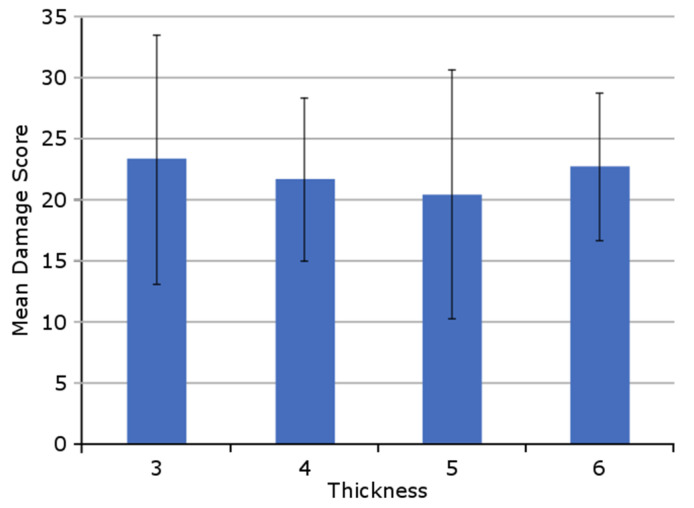
The distribution of the mean total damage score for the four inlay sizes (*n* = 47).

**Figure 5 materials-13-04589-f005:**
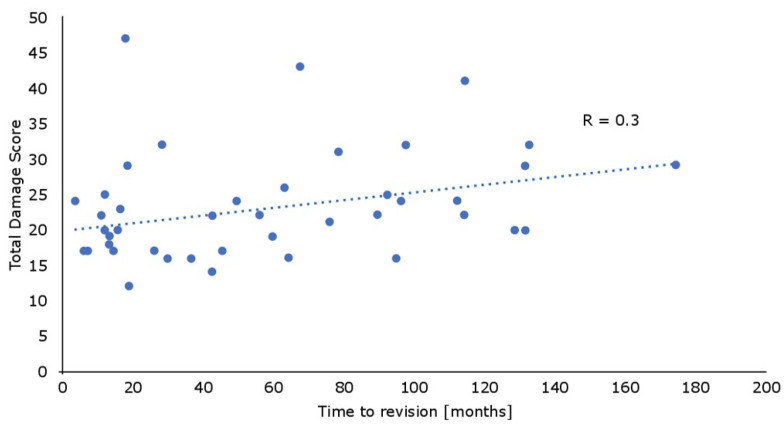
With the increasing time to revision, the total damage score increased (*p* < 0.05), but this correlation was weak (R = 0.3).

**Figure 6 materials-13-04589-f006:**
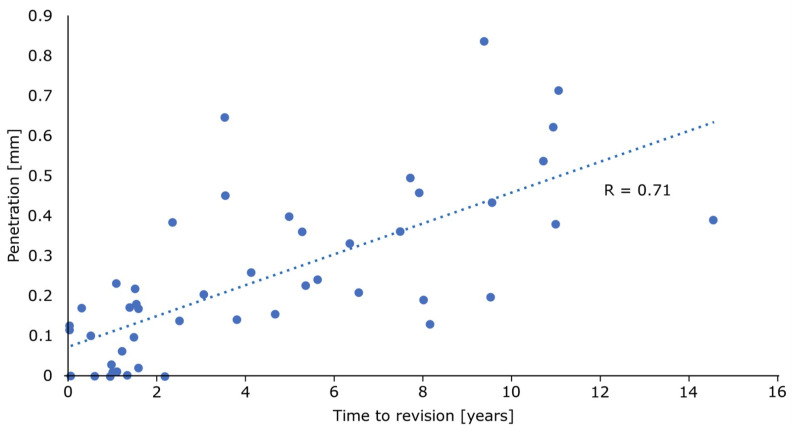
There is a strong correlation between the time to revision and penetration (R = 0.71; *p* < 0.001; *n* = 47).

**Figure 7 materials-13-04589-f007:**
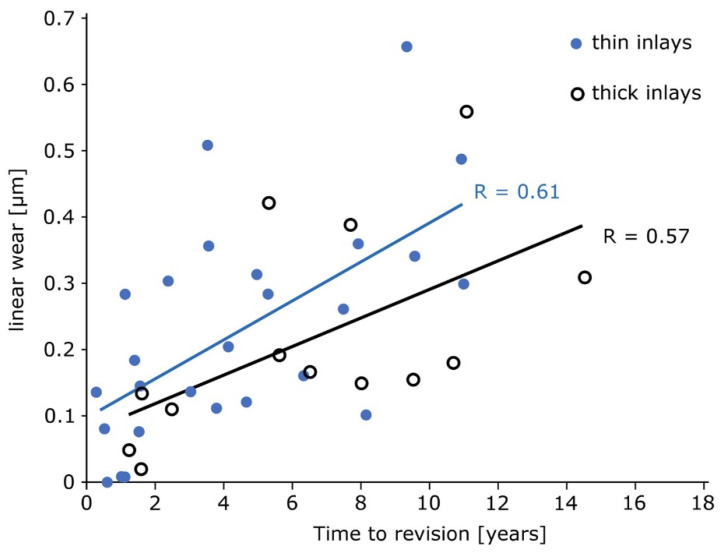
Linear wear shows comparable results for thin, as well as thick inlays (p = 0.173).

**Figure 8 materials-13-04589-f008:**
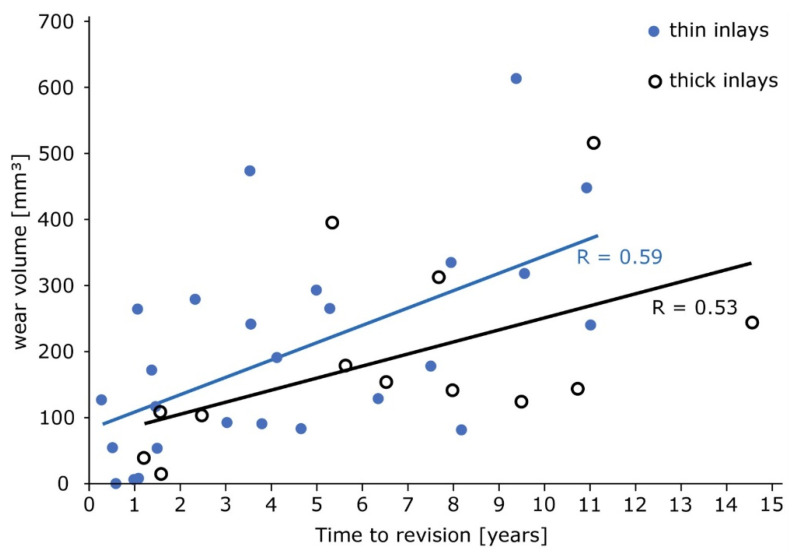
Thin and thick inlays show comparable wear volumes over time (p = 0.196).

**Figure 9 materials-13-04589-f009:**
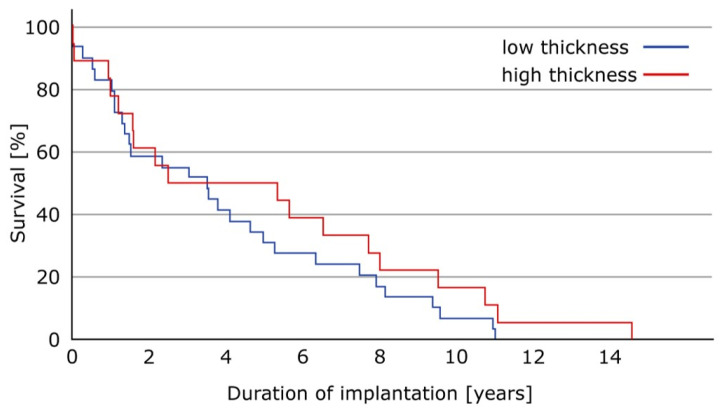
Kaplan-Meier blot for the two groups, showing no significant differences in survival (p = 0.264).

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
