# Peer review of "Bearing Thickness Is Not a Predictive Factor for Damage and Penetration in Oxford Unicompartmental Knee Arthroplasty—A Retrieval Analysis"

_materials, 2020, doi:10.3390/ma13204589_

Round 1

Reviewer 1 Report

The authors studied the effect of bearing thickness on clinical wear of the Oxford unicompartmental knee arthroplasty. Interestingly, they found that there was no thickness dependence of wear. The finding has novelty and it is recommended that the paper is published provided that the following comments are responded to.

Materials and Methods. Specify the type of UHMWPE, especially regarding possible irradiation dose, crosslinking, and method of sterilization.

Materials and Methods. The fact that TC (variance of thickness) is simply added to the nominal thickness needs to be better justified.

Results. Qualitative expressions such as “excellent agreement”, “weak correlation”, or “strong correlation” should be avoided. If the authors can find justifications for R values indicating weak or strong correlation, these should be provided.

Author Response

We would like to thank Reviewer 1 for his work, as it surely will improve our manuscript.

  1. With regards to the type of UHMWPE, the manufacturer changed the PE in 2005. In our cohort, 5 of the retrieved inlays were implanted before, the rest was implanted after the change. The UHMWPE used before 2005 was Hi-Fax 1900, the PE currently in use is GUR 1050. A publication by Mohammad and colleagues showed no significant differences in wear using a RSA analysis. We have added this information to the materials and methods section (l. 65ff). With regards to to other informations, we unfortunately were not able to obtain them.
  2. We have amended the respective paragraph as follows: “The mean variance of +0.130 mm was added, as the nominal thickness does not resemble the true thickness of a new inlay. We have found a mean variance of +0,130 mm in 10 brand new inlays, measured for this purpose. (l. 94ff)
  3. The mentioned expressions were changed in l. 121 (“excellent agreement” to “comparable results”) or omitted ll. 157-158. However, we disagree that these expression should be avoided at all costs, as these expression do put results in a certain perspective (see l. 138)

Reviewer 2 Report

Dear researchers, I congratulate you on your manuscript. This is an excellent manuscript but it needs some improvement for publication. First review your abstract to make it more explanatory and focus on the role of your manuscript. The introduction is very scarce, I think that the authors should justify their manuscript by adding epidemiological data. The material and methods are good, but the authors must introduce references so that the readers can reproduce. The results are well described, but the captions are incomplete and the figures should be more self-explanatory. References must be more current.

Author Response

We thank Reviewer 2 for the kind words and for the work he put in.

  1. With regards to the abstract, we feel it is concise and does explain the role of the manuscript fairly exact.
  2. We have added epidemiological data and changed the introduction accordingly (l. 32ff: […], the 15 year revision rate is around 11% (Mohammad et al. 2018).
  3. With regards to the reference in the materials and methods section, we have added newer references to the hood score (l. 77). Also, we have added an abstract with regards to the nominal and actual thickness of the inlays (l. 94ff). We have also added pictures of the types of damage, for better understanding of the types of damage (see figure 2). The figure of the digital gauge was removed – as asked by reviewer 3.
  4. We do not know how we should sensibly change the figures, since they are in our modest opinion fairly self-explanatory, as are the caption. We have, however replaced figure 2 (See above).
  5. As mentioned earlier, we added references to the introduction and the materials and methods.

Reviewer 3 Report

In this manuscript, Eckert et al present a thickness analysis of retrieved knee arthroplasty. They present data that indicates that the thickness of the bearing component is not a parameter that determines damage. The study is of clinical relevance, but the manuscript falls short in doing a detail report of the retrieved bearings.

For this reviewer, an important correlation would have been if the knees have be retrieved from right or left footed patients. Depending on the patient preference, higher mechanical demand is expected from one over the other knee.

A surface analysis is important, as a tribolayer forms during the initial sliding cycles. What was the surface roughness of the bearings after and the differences between them?

A conventional microscopy image showing the characteristic type of wear mechanisms (third body embedded, scratches, etc) needs to be included for clarity. Since this is a visual analysis, it is fundamental to share what was observed.

Figures 3 and 4 can be read easily, however, figures 5, 6, 7, 8, and 9 axis titles and values are very hard to read, inconsistent with other figures, and quality is poor.

The reviewer thinks that it is not needed to show the picture the digital gauge used for measuring the thickness, but rather, present some image of the retrieved bearings.

Author Response

We would like to thank reviewer 3, as his work does indeed help to improve the quality of our manuscript

  1. The reviewer remarked that the preferred foot would be of some importance. This is indeed a very interesting issue and we would agree that it absolute is worth evaluating. Richards and Higgins have analysed knee contact forces in patients with native knee with and without osteoarthritis and have found no significant differences in knee contact forces in healthy adults as well as adults with moderate osteoarthritis (Richards and Higgins, 2010 Knee Contact Force in Subjects with Symmetrical OA Grades: Differences between OA Severities). Also, in this cohort, given the retrospective nature of this retrieval study, we did not assess the contralateral side and do not have xrays and/or MRIs to evaluate the contralateral knees for osteoarthritis. We think that this would quite possibly play an even bigger role in mechanical demand, however, this cannot be assessed in a retrospective retrieval analysis.
  2. Conventional images showing characteristic types of wear have been added. (l. 100ff, figure 2).
  3. Similarly, a surface analysis cannot be conducted in a retrospective retrieval study. The surface can be very smooth if the main damage is burnishing or rather rough when scratching is the prevalent type of damage (see figure 2 in the amended manuscript). From our perspective, a roughness measurement in retrieval study will not be valid.
  4. The figures can be amended very easily and we can provide the publisher with vector graphics to suit any changes in size and spacing if needed, however, we were not able to upload said vector graphics or include them in the initial manuscript.
  5. See 2.

Round 2

Reviewer 3 Report

The authors did not address most of the comments previously reported by this reviewer.

Author Response

In the first review, reviewer 3 placed five points:

  • Side of the implants: retrieved from right or left footed patients. Depending on the patient preference, higher mechanical demand is expected from one over the other knee.
  • Surface analysis and surface roughness (measurement) of the bearings.
  • Conventional microscopy image showing the characteristic type of wear mechanisms.
  • Poor quality of figures.
  • Removing the picture the digital gauge.

Answers and comments:

We agreed on the last three comments and changed the manuscript accordingly.

3) Appropriate images (Fig.2) were already added in the first revision as suggested by the reviewer.

4) We offered to submit vector graphic files of higher resolution. However as the submission system does not support this solution, we have now increased the quality of the figures in the current word file. 

5) The picture has already been removed in the first revision as suggested by the reviewer.

Regarding the first two arguments (side of the knee and roughness measurements) we have a different opinion then reviewer 3 and we again want to explain why.

1) Side of the implants: retrieved from right or left footed patients. In fact mechanical loadings might be different, if the side of the knee is considered. However, it is not generally the side of the knee (in terms of left or right) that determines the loading condition; it is rather the question which is the dominant leg. For the upper extremities (like the arm or hand) most of the people know which their dominant side is and roughly 90% of the population is right-sided dominant in the upper extremities. However, for the lower extremities (as the leg, relevant in our study) it is unknown which side is the dominant for most of the people. For sure, competitive athletes might know the dominant leg but this is not the population that was studied in our cohort. Even me, as a professor of biomechanics, doing some sports, I don’t know exactly my dominant side of the leg.

Therefore from our point of view it is meaningless to correlate our wear results to the left or right knee. However, the dominant side is mostly unknown and is also not recorded in the clinical data and therefore this information is simply not available.  This is why we do not agree on the suggestion of reviewer 3.

2) Surface analysis and surface roughness (measurement) of the bearings. The clinical relevance of polyethylene wear is the amount of wear that is released to the human body causing proinflamatory reactions triggered by wear particles. In our study we have quantitatively assessed the amount of linear and volumetric wear. This is therefore the most important parameter from a clinical point of view. Regarding the surface of polyethylene components, the surface might be rough without releasing a huge amount of material (wear volume).  On the other side the surface might be smoothly worn with high amounts of released wear particles. From the topography of polyethylene components no direct information regarding the quantity of worn material can be gained. The topography may held to understand the acting wear mechanisms (like abrasion, adhesion etc.) but it does not help to get information on the clinical relevant amount of wear. In fact this might be different with metallic surfaces articulating against polyethylene, whereas a rougher metallic surface may increase the amount of removed polyethylene. However in this study metallic surfaces have not been investigated. Therefore we cannot follow the suggestion of reviewer 3 in this regard. We cannot see relevant additional information regarding the scope of our study if we would add the requested information. 

Moreover, we want to show that the requested information (side of the knee and surface roughness of the polyethylene) are not essential in terms of the current state of research and are not routinely performed in knee replacement retrieval analysis. We therefore listed three current publications on knee replacement retrieval analysis. None of these studies analyzed or included these parameters:

Cerquiglini, A. et al. Retrieval analysis of contemporary antioxidant polyethylene: multiple material and design changes may decrease implant performance. Knee Surgery, Sports Traumatology, Arthroscopy 27, 2111–2119 (2019).

Ponzio, D. Y. et al. Antioxidant-stabilized highly crosslinked polyethylene in total knee arthroplasty: a retrieval analysis. The Bone & Joint Journal 100-B, 1330–1335 (2018).

Spece, H. et al. Reasons for Revision, Oxidation, and Damage Mechanisms of Retrieved Vitamin E-Stabilized Highly Crosslinked Polyethylene in Total Knee Arthroplasty. The Journal of Arthroplasty 34, 3088–3093 (2019).